# Maternal body composition and gestational weight gain in relation to asthma control during pregnancy

**Danielle R. Stevens[1], Matthew C. H. Rohn[1], Stefanie N. Hinkle[1], Andrew D. Williams[2], Rajesh Kumar[3], Leah M. Lipsky[1], William Grobman[3], Seth Sherman[4], Jenna Kanner[1], Zhen Chen[1], Pauline Mendola[1,5]***

**1** Division of Intramural Population Health Research, *Eunice Kennedy Shriver* National Institute of Child Health and Human Development, Bethesda, MD, United States of America, **2** UND School of Medicine and Health Sciences, University of North Dakota, Grand Forks, ND, United States of America, **3** Northwestern University Feinberg School of Medicine, Chicago, IL, United States of America, **4** The Emmes Company, Rockville, MD, United States of America, **5** School of Public Health and Health Professions, University at Buffalo, Buffalo NY, United States of America

\* pmendola@buffalo.edu

**Data Availability Statement:** The analytic dataset is now available on BRADS, and is easily accessible with a proposal and DUA filled out as required by NICHD/DIPHR. BRADS datasets are routinely

## Abstract

### Background

Poor asthma control is common during pregnancy and contributes to adverse pregnancy outcomes. Identification of risk factors for poor gestational asthma control is crucial.

### Objective

Examine associations of body composition and gestational weight gain with asthma control in a prospective pregnancy cohort (n = 299).

### Methods

Exposures included pre-pregnancy body mass index (BMI), first trimester skinfolds, and tri-mester-specific gestational weight gain. Outcomes included percent predicted forced expiratory volumes (FEV1, FEV6), forced vital capacity (FVC), peak expiratory flow (PEF), FEV1/FVC, symptoms (activity limitation, nighttime symptoms, inhaler use, and respiratory symptoms), and exacerbations (asthma attacks, medical encounters). Linear and Poisson models examined associations with lung function ($\beta$ (95% confidence interval (CI)), asthma symptom burden (relative rate ratio (RR (95%CI)), and exacerbations (RR (95%CI)).

### Results

Women with a BMI $\geq$ 30 had lower percent predicted FVC across pregnancy ($\beta_{ThirdTrimester}$: -5.20 (-8.61, -1.78)) and more frequent night symptoms in the first trimester (RR: 1.66 (1.08, 2.56)). Higher first trimester skinfolds were associated with lower FEV1, FEV6, and FVC, and more frequent night symptoms and inhaler use across pregnancy. Excessive first tri-mester gestational weight gain was associated with more frequent activity limitation in the

released to outside investigators and this process has worked very well for all parties. All of the internal processes to add the data to the BRADS system are complete, and you should be able to view the instructions for data request at: https://brads.nichd.nih.gov/AccessRequest/AccessRequest/. You can view the new B-WELL-Mom webpage at: https://brads.nichd.nih.gov/OurCollections/BWell/.

**Funding:** This work was supported by the National Institutes of Health's Intramural Research Program at the Eunice Kennedy Shriver National Institute of Child Health and Human Development (clinical site contracts HHSN275201300013C to Northwestern University, HHSN275201300014C to the University of Alabama at Birmingham; and the Emmes Company for the Data Coordinating Center HHSN275201300026I, HHSN27500001, HHSN275000017). The funders had no role in study design, data collection and analysis, decision to publish, or preparation of the manuscript.

**Competing interests:** The authors have declared that no competing interests exist.

first trimester (RR: 3.36 (1.15, 9.80)) and inhaler use across pregnancy ($RR_{ThirdTrimester}$: 3.49 (1.21, 10.02)).

## Conclusions

Higher adiposity and first trimester excessive gestational weight gain were associated with restrictive changes in lung function and symptomology during pregnancy.

## Introduction

Asthma complicates approximately 10% of U.S. pregnancies, and has been associated with higher rates of adverse pregnancy outcomes [1–5]. In pregnancies affected by asthma, adequate asthma control may mitigate adverse pregnancy outcomes [6]. However, about a third of women experience deterioration in asthma control throughout pregnancy, and risk factors for inadequate asthma control during pregnancy have not been well-elucidated [7].

Evidence suggests that obesity and weight gain may be associated with changes in asthma development or control in non-pregnant populations [8–14]. Pre-pregnancy obesity and excess gestational weight gain may contribute to decrements in asthma control throughout pregnancy via mechanical restriction of lung volume, cardiometabolic disruption, and inflammation [15]. A few prior studies have examined associations between pre-pregnancy body mass index (BMI), gestational weight gain, and asthma exacerbations during pregnancy, with discordant results [16–20]. Further, asthma control is assessed based on exacerbation risk *and* symptomology; only one prior study has examined maternal pre-pregnancy BMI and gestational weight gain associated with asthma control based on symptomology during pregnancy [20, 21]. Contrary to hypotheses, this study reported that increased total gestational weight gain was associated with decreased risk of recurrent uncontrolled asthma [20].

These few prior studies also have several notable limitations. First, they frequently lack trimester-specific information, which may help guide clinical decision-making and knowledge of underlying biologic pathways. Second, prior studies do not include spirometry, despite its inclusion in many asthma control guidelines and its clinical utility [22]. Third, measures of body composition such as subscapular (central) or triceps (regional) skinfolds are more sensitive measures of body fatness than BMI [23]; prior studies lack these more sensitive measures. Finally, existing studies do not include a control group of women without asthma to assess whether associations are unique to women with asthma. Given the contrasting literature and limitations to prior research, further study of this association is needed.

The primary aim of this study was to examine maternal body composition and gestational weight gain in relation to gestational asthma control in the Breathe—Wellbeing, Environment, Lifestyle, and Lung Function (B-WELL-Mom) cohort. We hypothesized that higher adiposity (i.e., pre-pregnancy BMI and skinfold thicknesses) and excessive gestational weight gain would contribute to poorer lung function and more frequent asthma symptoms and exacerbations during pregnancy.

## Materials and methods

### Study design

Participants were part of the B-WELL-Mom study, a prospective pregnancy cohort of women with active asthma (n = 311) or no history of asthma (n = 107). Women with and without

asthma were recruited from two US-based study sites (Northwestern University in Chicago and the University of Alabama at Birmingham) during 2015–2019. Medical record review was used to identify potentially eligible participants, who were then screened for eligibility (S1 Table) and consent. Women were followed across pregnancy with three in-person clinical assessments of lung function and questionnaires completed at 6–14 weeks gestation (visit 1), 20–22 weeks gestation (visit 2), and 30–32 weeks gestation (visit 3). Study procedures also included daily diaries and medical record abstraction throughout pregnancy. Data collection was for research purposes only and was not used to direct clinical management of asthma. Study materials are available on the B-WELL-Mom website (https://b-well-mom.org) and in the supporting information (baseline questionnaire in S1 File, visits 2 and 3 questionnaire in S2 File, and daily diary questions in S3 File). Questionnaires were based on existing asthma control assessments [21, 24, 25] and were thus not validated prior to use on study participants. Human subjects' approval was obtained from all participating sites and all subjects provided written informed consent.

## Outcome assessment

According to asthma control guidelines, asthma control is typically assessed based on exacerbation risk and symptomology [21, 24, 26]. Lung function is not consistently employed across asthma control guidelines but plays a key role in asthma control assessment in clinical practice [21, 24, 26]. Thus, our asthma control outcomes included: 1) lung function, 2) symptomology, and 3) exacerbations.

Lung function (forced expiratory volume in 1 second (FEV1, percent predicted) and 6 seconds (FEV6, percent predicted), peak flow (percent predicted), forced vital capacity (FVC, percent predicted), and the ratio of FEV1 to FVC) was assessed via spirometry by trained study staff at each clinic visit, collected in triplicate, and the maximum value of each measure at that visit was selected for analyses. Additional measures of peak flow were obtained through daily diary report. Women were instructed in proper use and recording of the peak flow meter at study visit 1, and their technique was reviewed at each study visit. Spirometry protocol followed the National Health and Nutrition Examination Survey Respiratory Health Spirometry Procedures [27]. To account for variation in the timing and number of measurements per woman and to produce clinically interpretable measures by trimester, we calculated individual-level lung function at gestational weeks 14, 28, and delivery using unadjusted linear mixed models with a random intercept and gestational age slope.

Asthma symptom incidence was self-reported in daily diaries and included activity limitation (Did you miss school, work or normal daily activities because of your illness or symptoms?), nighttime symptoms (Did you wake due to difficulty breathing or coughing in the middle of the night?), rescue inhaler use (Did you take any prescription medications?), and respiratory symptoms (Did you experience any of the following symptoms: Wheezing, coughing, shortness of breath, chest tightness, chest pain?). If a woman reported having a cold, the flu, or a sore throat, we did not count symptoms for that day. We then calculated the cumulative incidence rate (days/trimester) of each asthma symptom per trimester based on the number of days that women reported experiencing each outcome per the total number of daily diaries recorded at gestational weeks 14, 28, and delivery.

We additionally examined associations with two measures of asthma exacerbations collected from visit questionnaires: asthma-related medical encounters and asthma attacks. Asthma-related medical encounters were calculated as the sum of reported times women visited a hospital, emergency department/urgent care center or had an unscheduled sick visit to their doctor due to asthma. Asthma attacks were calculated as the sum of reported times

women reported experiencing an asthma attack throughout pregnancy per the total number of days of pregnancy. In the first visit questionnaire, women reported asthma attacks and medical encounters from the past year; thus, attacks/encounters from the first study visit were estimated as a proportion of the total number reported from the past year (i.e., number of attacks/encounters x (number of gestational weeks at first study visit / 52 weeks) rounded to the nearest whole number).

## Exposure assessment

Maternal height, weight, subscapular skinfolds (mm), and triceps skinfolds (mm) were measured at each study visit by trained study staff following standardized study procedures. Skinfolds were measured using Lange skinfold Calipers (Beta Technology, Inc.) on the right side of the body, collected in triplicate, and averaged for analyses. First trimester sum of skinfolds was calculated as the sum of the subscapular and triceps skinfolds for analyses. Pre-pregnancy BMI (kg/m$^2$) was calculated based on self-reported pre-pregnancy weight and measured height at study visit 1 and categorized as BMI < 25, BMI 25–30, and BMI ≥ 30 [28]. Fifteen women with BMI < 18.5 were included in the BMI < 25 category for analyses.

Additional measures of weight were abstracted from medical records, providing between 2 and 23 (median: 14) weight measures throughout pregnancy. If women had more than 1 weight measure per gestational week, we calculated the mean for that week. To account for variation in the timing and number of measurements per woman and to produce clinically interpretable measures of gestational weight gain by trimester, we calculated individual-level gestational weight gain for each trimester using an unadjusted linear mixed spline model with knots at the end of the first and second trimesters, and a random intercept and slope for gestational age. We then calculated trimester-specific gestational weight gain adequacy based on the rate of gestational weight gain and the Institute of Medicine's guidelines (inadequate, adequate, excessive; adequate serves as the reference) [29].

## Covariates

Covariates for adjustment were selected a priori and included confounders collected by self-report at baseline, daily diaries, and study visits [30, 31]. These covariates included age (years), self-identified race/ethnicity (non-Hispanic White, non-Hispanic Black, Hispanic, Other), household income (quartiles: < $15,000, $15,000-$40,000, $40,000-$100,000, ≥$100,000), marital status (married and/or living with partner, divorced/separated or widowed, single), education (high school or less, Associate's/some college, Bachelor's degree, Master's or advanced degree), parity (nulliparous, multiparous), and pre-pregnancy cigarette smoke exposure (yes/no). Models for gestational weight gain additionally adjusted for BMI, diabetes, and hypertension. All models were additionally adjusted for study site. In sensitivity analyses, we additionally controlled for baseline asthma medication regimen (step 1, step 2, step 3, or step 4+, see S2 Table for definitions) and baseline asthma control according to the Asthma Control Test at visit 1 (well-controlled (score > 19) or poorly-controlled (score ≤ 19)) [25].

## Statistical analyses

The aim of this study was to examine maternal body composition and gestational weight gain in relation to asthma control during pregnancy. Multivariable linear regression models examined associations of BMI, skinfolds, and gestational weight gain with mean (95% confidence intervals (CI)) lung function at the end of each trimester. Comparisons are presented as β (95% CI). Multivariable Poisson regression models examined associations of BMI, skinfolds, and gestational weight gain with incidence (days/trimester (95% CI)) of asthma symptoms at

the end of each trimester. Comparisons are presented as relative rate ratios (RR (95% CI)). Poisson models included an offset for the number of daily diaries women had completed. Models were run separately for each exposure and each trimester. Models in the second and third trimesters included adjustment for gestational weight gain in prior trimesters.

Asthma exacerbations occurred too infrequently to calculate trimester-specific associations. However, Poisson regression models examined the overall association of each exposure with the number of asthma attacks and medical encounters during pregnancy. These models included an offset for gestational age at delivery.

In secondary analyses, models from primary analyses were re-run to examine associations between all exposures and individual respiratory symptoms (i.e., rate of wheezing, shortness of breath, coughing, chest tightness, or chest pain) and individual exacerbations (i.e., hospitalizations, emergency department/urgent care center, or sick visit to doctor) as our outcomes.

We examine the robustness of our results through two sensitivity analyses. First, we re-ran our primary analysis including additional adjustments for asthma control and medication regimen at baseline. Second, we repeated our primary analysis among women without asthma in order to assess whether our associations may be due to mechanisms that are not asthma-specific (e.g. mechanical restrictions in lung volume).

Missing BMI (n = 58, 19.4%) and cigarette smoking (n = 1, 0.3%) were imputed using 10 multiple chained equations and included all covariates from our primary analysis as well as employment and student status, health insurance, receipt of federal aid (e.g., social security recipient), asthma control variables from the asthma control test, asthma medication regimen, highest recalled weight before pregnancy, lowest recalled weight before pregnancy, pre-pregnancy somatotype, and visit 1 weight. Participant characteristics by BMI category (BMI < 25, BMI 25–30, BMI ≥ 30, missing BMI) are presented in S2 Table.

SAS version 9.4 (Cary, North Carolina, US) and R version 3.6.1 (Vienna, Austria) were used for analyses, and $p \leq 0.05$ was used to determine statistical significance. No adjustments were made for multiple comparisons [32].

## Results and discussion

Primary analyses were limited to 299 women with asthma, and sensitivity analyses included 101 women without asthma (Fig 1). Women with asthma were on average 29.7 (standard deviation: 5.9) years of age at enrollment and predominantly non-Hispanic Black (n = 157 (52.5%)). Most women with asthma were on a step 1 asthma medication regimen (n = 157 (52.5%)) and had poorly-controlled asthma at enrollment (n = 154 (51.5%)).

Mean estimated lung function by pre-pregnancy BMI and gestational weight gain per trimester are presented in Fig 2. Compared to women with BMI < 25, those with BMI ≥ 30 had an approximately 5-unit lower percent predicted FVC across all trimesters of pregnancy (S3 Table). Compared to women with adequate second trimester gestational weight gain, excessive second trimester gestational weight gain was associated with higher second and third trimester percent predicted PEF ($\beta_{\text{second trimester}}$: 6.98 (0.42, 13.54), $\beta_{\text{third trimester}}$: 8.78 (0.62, 16.94)). First trimester skinfolds–notably subscapular skinfolds–were negatively associated with percent predicted FEV1, FEV6, and FVC (Table 1).

Estimated incidence of asthma symptoms by pre-pregnancy BMI and gestational weight gain per trimester are presented in Fig 3. Compared to women with BMI < 25, those with BMI ≥ 30 had more days with night symptoms in the first trimester (RR: 1.66 (1.08, 2.56)) (S4 Table).

Compared to women with adequate first trimester gestational weight gain, inadequate and excess first trimester gestational weight gain were associated with more days with activity

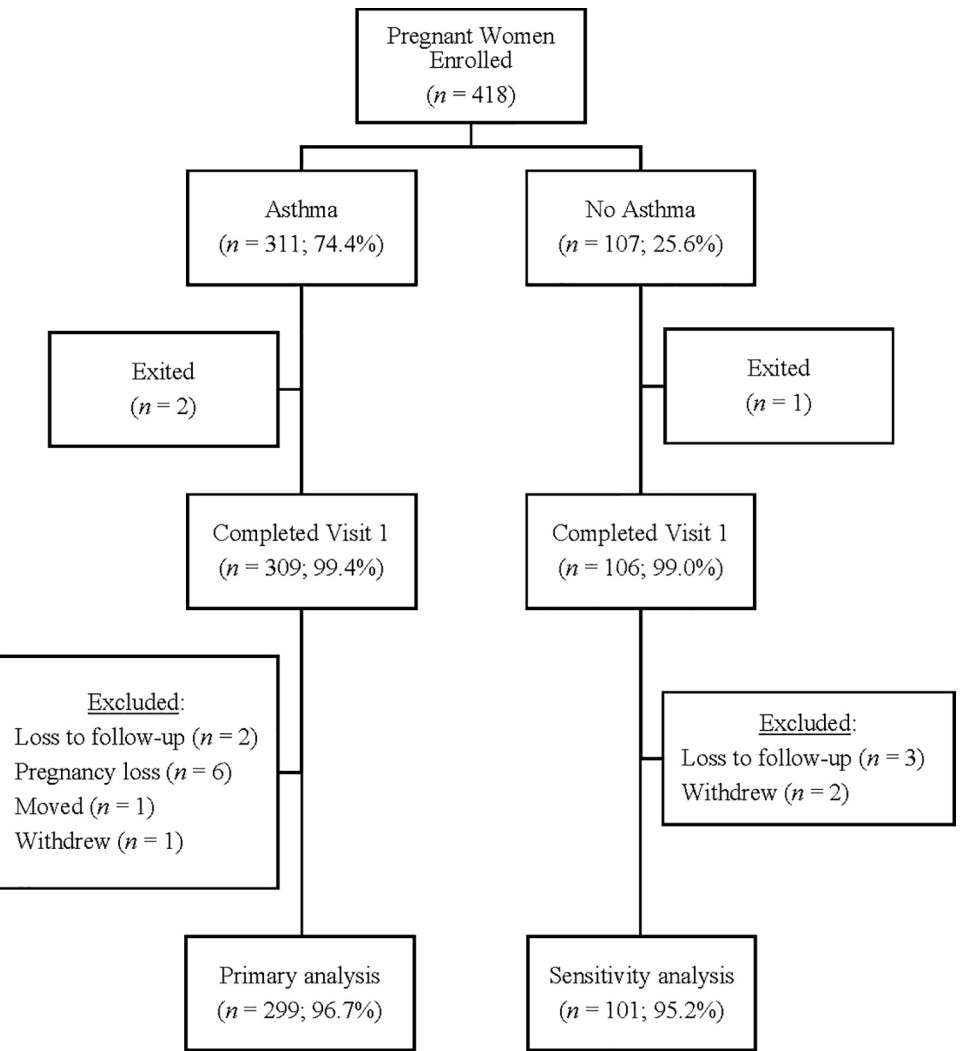

**Fig 1. Flow diagram of study population for women in the Breathe-Wellbeing, Environment, Lifestyle, and Lung Function Study, 2015–2019, USA.**

limitation in the first trimester (RR: 3.73 (1.18, 11.78), RR: 3.36 (1.15, 9.80), respectively). Excess first trimester gestational weight gain was also associated with more days with rescue inhaler use across all trimesters ($RR_{first\ trimester}$: 2.57 (1.01, 6.51), $RR_{second\ trimester}$: 2.89 (1.13, 7.41), $RR_{third\ trimester}$: 3.49 (1.21, 10.02)). Inadequate first trimester gestational weight gain was associated with fewer days with respiratory symptoms in the first trimester (RR: 0.63 (0.42, 0.95)). First trimester triceps skinfolds and sum of skinfolds were positively associated with activity limitation, night symptoms, and rescue inhaler use (Table 2). Results from secondary analyses for individual respiratory symptoms indicated that first trimester skinfolds were positively associated with wheeze and cough, and inadequate first trimester gestational weight gain was associated with fewer days of chest tightness across pregnancy (S5 Table).

Estimated incidence of asthma exacerbations during pregnancy by pre-pregnancy BMI and gestational weight gain are presented in Fig 4. Compared to women with a BMI < 25, women with a BMI 25–30 had fewer asthma-related medical encounters during pregnancy (RR: 0.41 (0.21, 0.82)) (S6 Table). Results from secondary analyses for individual asthma exacerbations

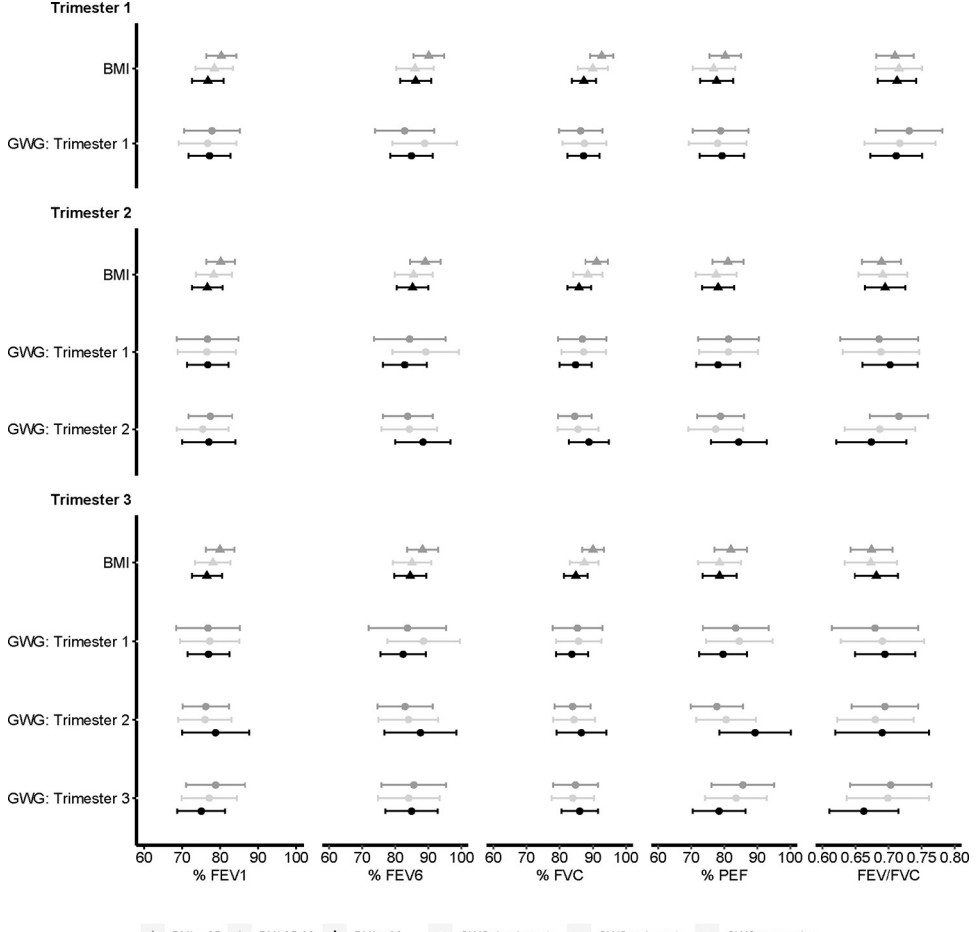

**Fig 2. Estimated means and 95% confidence intervals for lung function per trimester by pre-pregnancy BMI and GWG in the Breathe–Wellbeing, Environment, Lifestyle, and Lung Function Study, 2015–2019, USA.**
Abbreviations: % FEV1, percent predicted forced expiratory volume in 1 second; % FEV6, percent predicted forced expiratory volume in 6 seconds; % FVC, percent predicted forced vital capacity; % PEF, percent predicted peak flow; BMI, body mass index; FEV1/FVC, ratio of forced expiratory volume in 1 second to forced vital capacity; GWG, gestational weight gain.

indicated that a BMI 25–30 and BMI ≥ 30 were associated with decreased doctor's visits compared to a BMI < 25 (RR: 0.18 (0.06, 0.53), RR: 0.47 (0.28, 0.79), respectively). First trimester subscapular skinfolds were positively associated with emergency department/urgent care visits (RR: 1.43 (1.05, 1.94)) and negatively associated with doctor's visits (RR: 0.71 (0.52, 0.97)). Compared to adequate third trimester gestational weight gain, excessive third trimester gestational weight gain was associated with increased emergency department/urgent care visits (RR: 3.08 (1.12, 8.48)).

After additional adjustment for baseline asthma control and medication use, results for lung function were similar to our primary analysis, and effects were attenuated for asthma symptom outcomes (S7–S9 Tables). Among women without asthma, we did not observe many statistically significant associations, nor were effect sizes similar to analyses among women with asthma (S10 and S11 Tables).

In this prospective cohort study, we report on associations of maternal body composition and gestational weight gain with asthma control during pregnancy. Our findings suggest that

**Table 1. Adjusted[a] association for an IQR-increase in maternal body composition associated with lung function in the Breathe-Wellbeing, Environment, Lifestyle, and Lung Function Study, 2015–2019, USA.**

| | IQR[b] | % FEV1 | | % FEV6 | | % FVC | | % PEF | | FEV1/FVC | |
|---|---|---|---|---|---|---|---|---|---|---|---|
| | | β | 95% CI | β | 95% CI | β | 95% CI | β | 95% CI | β | 95% CI |
| First trimester | | | | | | | | | | | |
| Subscapular skinfold | 13.0 | **-3.49** | **-5.75, -1.22** | **-3.28** | **-5.78, -0.78** | **-3.93** | **-5.91, -1.94** | 0.39 | -2.36, 3.14 | -0.003 | -0.019, 0.014 |
| Triceps skinfold | 13.0 | **-1.98** | **-4.32, 0.36** | -2.24 | -4.97, 0.50 | **-3.19** | **-5.24, -1.15** | -0.20 | -3.01, 2.61 | 0.009 | -0.008, 0.025 |
| Sum of skinfolds | 22.5 | **-2.85** | **-5.03, -0.68** | **-2.84** | **-5.29, -0.40** | **-3.70** | **-5.60, -1.80** | 0.11 | -2.52, 2.74 | 0.003 | -0.013, 0.019 |
| Second trimester | | | | | | | | | | | |
| Subscapular skinfold | 13.0 | **-3.36** | **-5.53, -1.18** | **-3.25** | **-5.77, -0.73** | **-3.85** | **-5.79, -1.90** | 0.20 | -2.48, 2.87 | -0.002 | -0.019, 0.015 |
| Triceps skinfold | 13.0 | **-2.10** | **-4.35, 0.14** | -2.10 | -4.86, 0.66 | **-3.13** | **-5.14, -1.12** | -0.43 | -3.16, 2.29 | 0.007 | -0.011, 0.024 |
| Sum of skinfolds | 22.5 | **-2.85** | **-4.94, -0.75** | -2.76 | -5.23, -0.30 | **-3.63** | **-5.49, -1.76** | -0.12 | -2.67, 2.44 | 0.002 | -0.014, 0.019 |
| Third trimester | | | | | | | | | | | |
| Subscapular skinfold | 13.0 | **-3.27** | **-5.42, -1.12** | **-3.22** | **-5.78, -0.66** | **-3.75** | **-5.66, -1.84** | -0.18 | -3.01, 2.65 | 0.000 | -0.018, 0.018 |
| Triceps skinfold | 13.0 | **-2.20** | **-4.42, 0.02** | -1.97 | -4.78, 0.83 | **-3.04** | **-5.02, -1.07** | -0.82 | -3.71, 2.06 | 0.006 | -0.013, 0.024 |
| Sum of skinfolds | 22.5 | **-2.85** | **-4.91, -0.78** | **-2.69** | **-5.20, -0.18** | **-3.53** | **-5.36, -1.70** | -0.51 | -3.22, 2.19 | 0.003 | -0.014, 0.020 |

*Abbreviations*: % FEV1, percent predicted forced expiratory volume in 1 second; % FEV6, percent predicted forced expiratory volume in 6 seconds; % FVC, percent predicted forced vital capacity; %PEF, percent predicted peak flow; CI, confidence interval; FEV1/FVC, ratio of forced expiratory volume in 1 second to forced vital capacity; IQR, interquartile range

Bold represents statistically significant ($p \leq 0.05$) findings

[a]Models were adjusted for study site, age, race/ethnicity, household income, marital status, education, parity, and pre-pregnancy cigarette smoke exposure. Models for gestational weight gain were additionally adjusted for pre-pregnancy BMI, diabetes, and hypertension.

[b]Units are in millimeters.

women with asthma and higher adiposity (i.e., pre-pregnancy BMI and skinfold thicknesses) were more likely to have a restrictive pattern on spirometry and poor asthma control–as assessed by symptomology and emergency department/urgent care visits–during pregnancy. Gestational weight gain was not related to lung function, but excess gestational weight gain in the first trimester was positively associated with symptomology and excess gestational weight gain in the third trimester was positively associated with emergency department/urgent care visits. These results were consistent across trimesters and similar even with additional adjustment for baseline asthma control and medication use. Notably, these associations were not observed among women without asthma.

Few studies have examined whether pre-pregnancy BMI and gestational weight gain influence asthma control in pregnant populations. Most prior studies have focused only on asthma exacerbations and have observed inconsistent findings. Two studies have reported positive associations [17, 19] and two studies have reported no associations [20, 33] between pre-pregnancy BMI and asthma exacerbations during pregnancy. Similar to our study, Ali *et al* found that a higher pre-pregnancy BMI was associated with fewer overall asthma exacerbations [16]. In our study, this finding may be due to the strong negative association between maternal obesity and mild asthma exacerbations (i.e., asthma-related doctor's visits). The obese asthma phenotype is generally characterized by a more severe disease course [10]; thus, women with increased adiposity may be accustomed to poor asthma control and less likely to seek primary care assistance with their asthma during pregnancy. Importantly, this lack of managed care may have contributed to our findings of poor asthma control in these women. Results for severe asthma exacerbations reveal that higher maternal body composition and excessive third trimester gestational weight gain were associated with more emergency department/urgent care visits. In the literature, gestational weight gain has been positively [16], negatively [33],

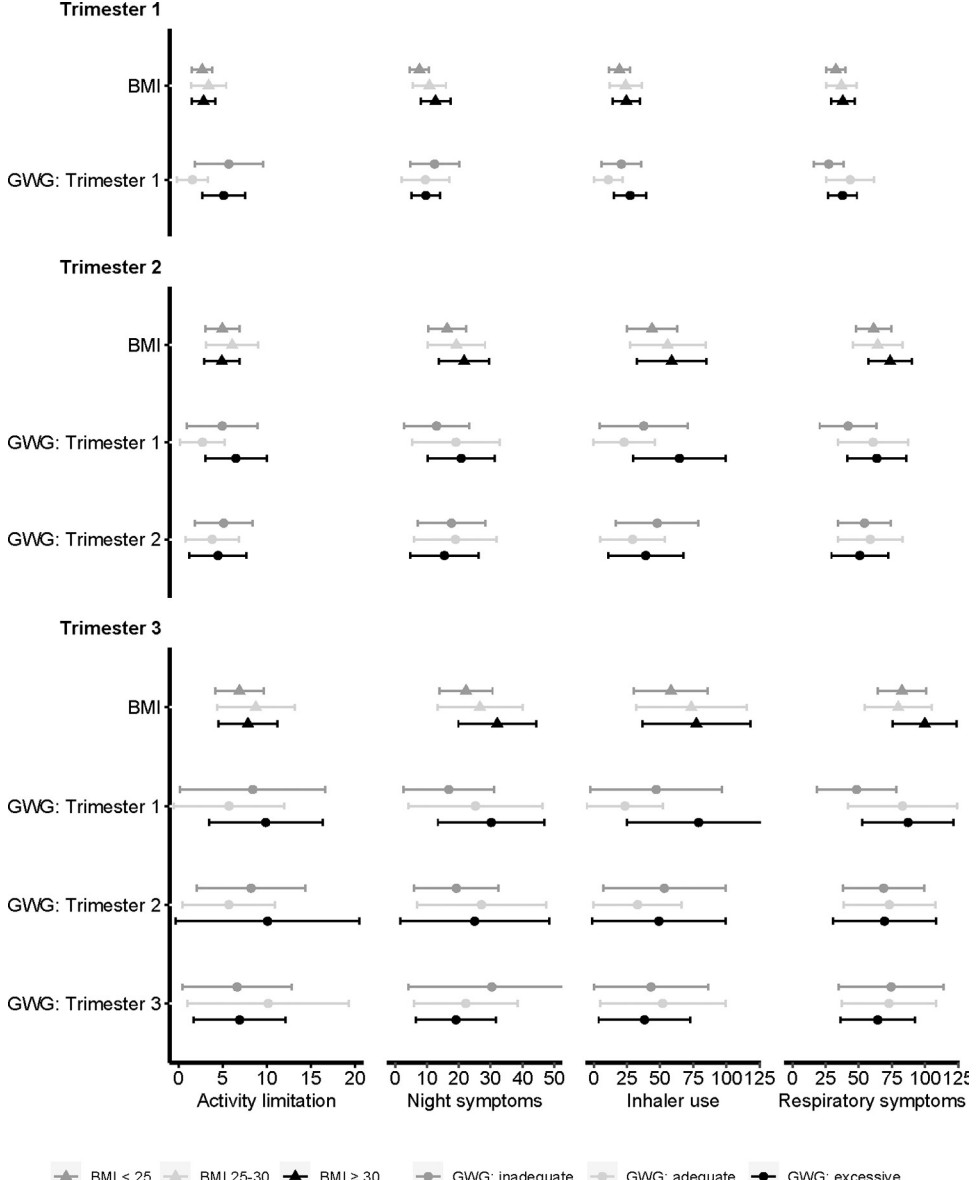

**Fig 3. Estimated mean days and 95% confidence intervals of asthma symptom burden per trimester by pre-pregnancy BMI and GWG in the Breathe-Wellbeing, Environment, Lifestyle, and Lung Function Study, 2015–2019, USA.** Abbreviations: BMI, body mass index; GWG, gestational weight gain.

and not associated with asthma exacerbations [17, 20]. Few studies have examined associations for trimester-specific gestational weight gain. Discrepancies in the literature may be due to a number of factors including differences in study populations, definitions of asthma exacerbations, and modelling of gestational weight gain. Given inconsistencies in prior research and the novelty of our findings, replication is needed.

To our knowledge, only one other study has examined associations of maternal body composition and gestational weight gain with asthma control based on symptomology during pregnancy. This study found no associations for pre-pregnancy BMI, and that total gestational weight gain was associated with decreased risk of recurrent uncontrolled asthma [20].

**Table 2. Adjusted[a] association for an IQR-increase in maternal body composition associated with incidence of asthma symptoms in the Breathe-Wellbeing, Environment, Lifestyle, and Lung Function Study, 2015–2019, USA.**

|  | IQR[b] | Activity limitation[c] | | Night symptoms[c] | | Rescue inhaler use[c] | | Respiratory symptoms[c] | |
|---|---|---|---|---|---|---|---|---|---|
|  |  | RR | 95% CI | RR | 95% CI | RR | 95% CI | RR | 95% CI |
| First trimester |  |  |  |  |  |  |  |  |  |
| Subscapular skinfold | 13.0 | 1.10 | 0.84, 1.43 | 1.16 | 0.92, 1.47 | 1.01 | 0.81, 1.25 | 1.04 | 0.91, 1.19 |
| Triceps skinfold | 13.0 | **1.60** | **1.23, 2.06** | **1.39** | **1.11, 1.73** | **1.38** | **1.12, 1.71** | 1.10 | 0.96, 1.27 |
| Sum of skinfolds | 22.5 | **1.33** | **1.04, 1.69** | **1.29** | **1.04, 1.59** | 1.18 | 0.97, 1.45 | 1.07 | 0.94, 1.22 |
| Second trimester |  |  |  |  |  |  |  |  |  |
| Subscapular skinfold | 13.0 | 1.08 | 0.87, 1.34 | 1.22 | 0.98, 1.51 | 1.07 | 0.85, 1.34 | 1.08 | 0.95, 1.23 |
| Triceps skinfold | 13.0 | 1.13 | 0.90, 1.43 | 1.20 | 0.97, 1.47 | **1.40** | **1.12, 1.75** | 1.07 | 0.94, 1.22 |
| Sum of skinfolds | 22.5 | 1.11 | 0.90, 1.36 | **1.22** | **1.00, 1.49** | **1.23** | **1.00, 1.53** | 1.08 | 0.96, 1.22 |
| Third trimester |  |  |  |  |  |  |  |  |  |
| Subscapular skinfold | 13.0 | 1.16 | 0.94, 1.44 | **1.29** | **1.04, 1.59** | 1.10 | 0.87, 1.38 | 1.06 | 0.93, 1.21 |
| Triceps skinfold | 13.0 | 1.17 | 0.92, 1.48 | **1.27** | **1.04, 1.56** | **1.41** | **1.12, 1.78** | 1.07 | 0.93, 1.22 |
| Sum of skinfolds | 22.5 | 1.17 | 0.95, 1.44 | **1.30** | **1.07, 1.58** | **1.26** | **1.01, 1.57** | 1.07 | 0.94, 1.21 |

*Abbreviations: CI, confidence interval; IQR, interquartile range; RR, relative rate ratio*

*Bold represents statistically significant (p ≤ 0.05) findings*

*[a]Models were adjusted for study site, age, race/ethnicity, household income, marital status, education, parity, and pre-pregnancy cigarette smoke exposure. Models for gestational weight gain were additionally adjusted for pre-pregnancy BMI, diabetes, and hypertension.*

*[b]Units are in millimeters*

*[c]The number of days women reported experiencing that outcome in daily diaries per the total number of daily diaries per trimester.*

Similarly, we report few associations between pre-pregnancy BMI and asthma symptoms, but that gestational weight gain–in particular excessive first trimester gestational weight gain–was associated with more days of activity limitation and rescue inhaler use. Thus, first trimester may be a critical period during which asthma control during pregnancy may be modified. Unexpectedly, we observed few associations between gestational weight gain and lung function other than improved percent predicted peak flow with excessive second trimester gestational weight gain. Given the lack of biologic plausibility, this finding may be due to chance. Increased rescue inhaler use may therefore not be due to a worsening of lung function; rather, it may represent an increased perception of dyspnea and fatigue associated with pregnancy-related weight gain. Lung function during pregnancy has not been well-described in asthmatic populations. In 2020, Jensen *et al* reported that women with asthma experienced reduced percent predicted FEV1, percent predicted FVC, and FEV/FVC, which decreased with advancing gestation [34]. In line with our findings, this study reported that higher pre-pregnancy BMI–but not gestational weight gain–was associated with poorer lung function in women with asthma during pregnancy. Given the consistency between our findings and previous studies, it appears that maternal obesity exerts a more notable influence on lung function, and excess gestational weight gain exerts a more notable influence on symptomology, during pregnancy. The current literature linking obesity and asthma does not suggest that the method used to induce changes in weight (i.e., diet, exercise, and/or bariatric surgery) matters more than the extent of the weight change when it comes to influencing asthma control [35]. Nevertheless, the potential role of lifestyle factors and other mechanisms (e.g., weight gain-related pregnancy complications) underlying these associations bear further study.

Our study includes a novel exposure of first trimester skinfold thicknesses which assesses central (subscapular) and regional (triceps) adiposity. We note strong negative associations between higher subscapular skinfolds with poorer lung function, and triceps skinfolds with

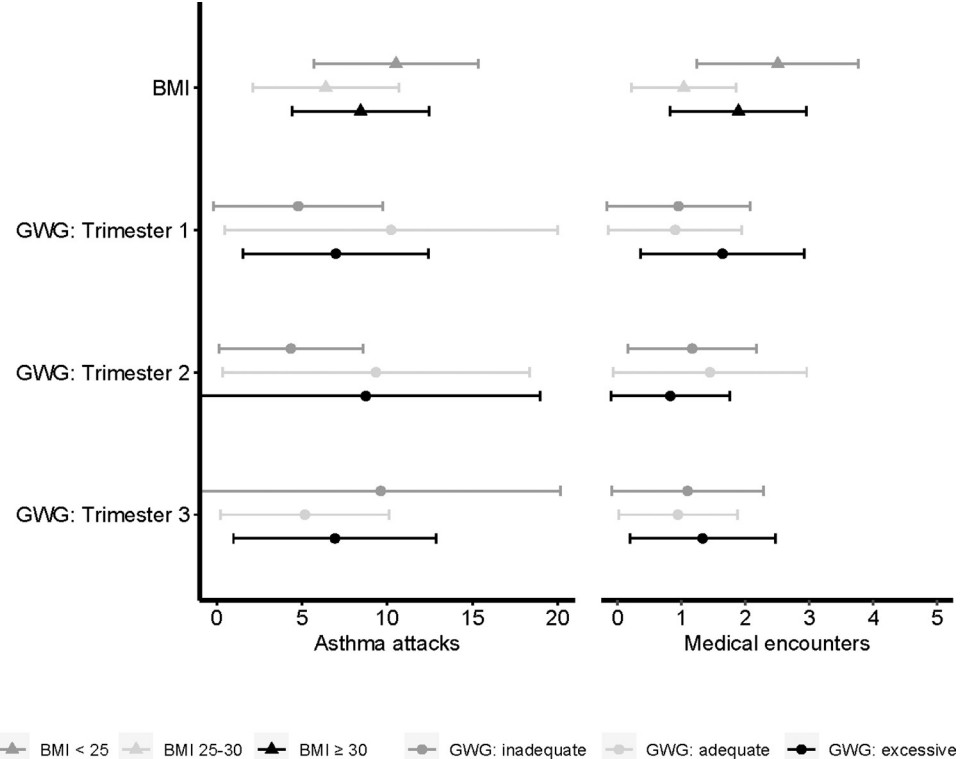

**Fig 4. Estimated mean number and 95% confidence intervals for asthma exacerbations during pregnancy by pre-pregnancy BMI and GWG in the Breathe-Wellbeing, Environment, Lifestyle, and Lung Function Study, 2015–2019, USA.** Abbreviations: BMI, body mass index; GWG, gestational weight gain.

asthma symptoms, suggesting body composition may be differentially related to different asthma control outcomes. Central adiposity may be linked to reduced lung function through mechanical restrictions to lung volume whereas regional adiposity may be linked to asthma symptoms through inflammatory or cardiometabolic pathways. In non-pregnant populations, skinfolds are reliable measures of body composition [23] and have been associated with poorer lung function [36]. During pregnancy, skinfold assessments to determine body composition is highly debated due to the extensive modifications the maternal unit undergoes during this time [37, 38]. However, skinfolds are commonly used to estimate body composition during pregnancy, are useful in research and clinical settings, and are associated with neonatal outcomes [37–39].

Our study also includes a novel sensitivity analyses conducted among women without asthma. These analyses found few differences in lung function measures by body composition or gestational weight gain, though percent predicted FEV6 was lower among women with higher first trimester subscapular skinfolds. This finding is in line with prior studies, which report that pre-pregnancy BMI and gestational weight gain is not associated with differences in lung function across pregnancy in healthy populations [34, 40]. Difference in associations between women with and without asthma may be attributed to differences in sample sizes in these groups; however, effect sizes were also dissimilar. Overall, the contrast between findings among women with and without asthma suggests that mechanical restriction of lung volume or cardiometabolic pathways associated with excess adiposity may not be solely responsible for our observed associations. Rather, other mechanisms may be operating to influence asthma control and restrictive lung function among women with asthma.

## Limitations

This study was limited by a small sample size which precluded our ability to conduct subgroup analyses by asthma phenotypes. Participant self-report of certain exposure (pre-pregnancy BMI) or outcome (symptomology, exacerbations) may be subject to measurement error. Self-reported pre-pregnancy weights were highly correlated with first measured pregnancy weights (n = 343, r = 0.98) as well as chart-abstracted pre-pregnancy weights (n = 118, r = 0.94) in our cohort, and missing data imputation for pre-pregnancy BMI allowed us to produce unbiased and efficient estimates for our associations [41]. Symptomology was self-reported in daily diaries, which have been shown to provide more sensitive estimates of asthma control than retrospective questionnaires [42]. Exacerbations–especially for the first trimester, which encompassed a one-year timeframe–were subject to recall bias. However, given the severity of asthma exacerbations and disruption that they would cause, any recall bias of exacerbations is likely limited. Finally, though we account for chronic mediation use at baseline, we did not assess how changes in medication use may have impacted our associations [43].

## Strengths

This is a prospective pregnancy cohort with comprehensive data ascertainment across pregnancy in women with and without asthma. Our study expands on prior studies in four key ways. First, we included objective and subjective measures of asthma control including lung function, as well as asthma symptoms. Second, we produced clinically interpretable results for each trimester of pregnancy. Third, we examined associations using body composition in addition to pre-pregnancy BMI [44]. Finally, we included women without asthma to assess whether associations differed in this subgroup.

## Conclusions

Poor asthma control affects 57–80% of patients [45] and is modified for more than half of women during pregnancy [7], making identification of risk factors a priority. In this prospective pregnancy cohort of women with and without asthma, we observed asthma-specific associations between higher pre-pregnancy BMI, early pregnancy skinfolds, and first trimester excessive gestational weight gain associated with restrictive changes in lung function and poor asthma control during pregnancy. Despite experiencing decrements in lung function and asthma control, women with an obese asthma phenotype had fewer asthma-related doctor's visits, suggesting less active management of asthma in this high-risk group. Active asthma management is key to the control of asthma and mitigation of adverse outcomes in pregnancies complicated by asthma [6]. While prenatal care visits are unrelated to asthma, they may provide an opportunity to help pregnant persons with asthma–particularly high-risk individuals with obesity or excess first trimester gestational weight gain–achieve better asthma control.

## Supporting information

**S1 Table. Eligibility and exclusion criteria in the Breathe-Wellbeing, Environment, Lifestyle, and Lung Function Study, 2015–2019, USA.**
(DOCX)

**S2 Table. Definitions used to categorize women according to asthma medication regimen in the Breathe-Wellbeing, Environment, Lifestyle, and Lung Function Study, 2015–2019, USA.**
(DOCX)

**S3 Table. Participant characteristics among women with asthma by pre-pregnancy BMI in the Breathe-Wellbeing, Environment, Lifestyle, and Lung Function Study, 2015–2019, USA.**
(DOCX)

**S4 Table. Adjusted[a] association of maternal pre-pregnancy BMI and gestational weight gain with lung function in the Breathe-Wellbeing, Environment, Lifestyle, and Lung Function Study, 2015–2019, USA.**
(DOCX)

**S5 Table. Adjusted[a] association between maternal pre-pregnancy BMI and gestational weight gain with incidence of asthma symptom in the Breathe-Wellbeing, Environment, Lifestyle, and Lung Function Study, 2015–2019, USA.**
(DOCX)

**S6 Table. Adjusted[a] association between maternal body composition and gestational weight gain with individual respiratory symptoms in the Breathe-Wellbeing, Environment, Lifestyle, and Lung Function Study, 2015–2019, USA.**
(DOCX)

**S7 Table. Adjusted[a] association of maternal body composition and gestational weight gain with asthma exacerbations across pregnancy in the Breathe-Wellbeing, Environment, Lifestyle, and Lung Function Study, 2015–2019, USA.**
(DOCX)

**S8 Table. Sensitivity analyses for adjusted[a] association of maternal pre-pregnancy BMI and gestational weight gain with lung function in the Breathe-Wellbeing, Environment, Lifestyle, and Lung Function Study, 2015–2019, USA.**
(DOCX)

**S9 Table. Sensitivity analyses for adjusted[a] association between maternal pre-pregnancy BMI and gestational weight gain with incidence of asthma symptom in the Breathe-Wellbeing, Environment, Lifestyle, and Lung Function Study, 2015–2019, USA.**
(DOCX)

**S10 Table. Sensitivity analyses for adjusted[a] association of maternal body composition and gestational weight gain with asthma exacerbations across pregnancy in the Breathe-Wellbeing, Environment, Lifestyle, and Lung Function Study, 2015–2019, USA.**
(DOCX)

**S11 Table. Sensitivity analyses for adjusted[a] association of maternal pre-pregnancy BMI and gestational weight gain with lung function among women without asthma in the Breathe-Wellbeing, Environment, Lifestyle, and Lung Function Study, 2015–2019, USA.**
(DOCX)

**S12 Table. Sensitivity analyses for adjusted[a] association between maternal pre-pregnancy BMI and gestational weight gain with incidence of asthma-like symptom among women without asthma in the Breathe-Wellbeing, Environment, Lifestyle, and Lung Function Study, 2015–2019, USA.**
(DOCX)

**S1 File.**
(PDF)

**S2 File.**
(PDF)

**S3 File.**
(PDF)

## Author Contributions

**Conceptualization:** Danielle R. Stevens, Matthew C. H. Rohn, Rajesh Kumar, Leah M. Lipsky, William Grobman, Jenna Kanner, Pauline Mendola.

**Data curation:** Danielle R. Stevens, Seth Sherman, Zhen Chen, Pauline Mendola.

**Formal analysis:** Danielle R. Stevens, Stefanie N. Hinkle, Andrew D. Williams, Seth Sherman, Zhen Chen, Pauline Mendola.

**Funding acquisition:** Pauline Mendola.

**Investigation:** Rajesh Kumar, Leah M. Lipsky, William Grobman, Seth Sherman, Zhen Chen, Pauline Mendola.

**Methodology:** Danielle R. Stevens, Matthew C. H. Rohn, Andrew D. Williams, Rajesh Kumar, Leah M. Lipsky, William Grobman, Jenna Kanner, Pauline Mendola.

**Resources:** Pauline Mendola.

**Writing – original draft:** Danielle R. Stevens.

**Writing – review & editing:** Danielle R. Stevens, Matthew C. H. Rohn, Stefanie N. Hinkle, Andrew D. Williams, Rajesh Kumar, Leah M. Lipsky, William Grobman, Seth Sherman, Jenna Kanner, Zhen Chen, Pauline Mendola.

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
