## [Decision Letter · Decision Letter 0]

5 Nov 2021

PONE-D-21-21560Maternal body composition and gestational weight gain in relation to asthma control during pregnancyPLOS ONE

Dear Dr. Mendola,

Thank you for submitting your manuscript to PLOS ONE. After careful consideration, we feel that it has merit but does not fully meet PLOS ONE’s publication criteria as it currently stands. Therefore, we invite you to submit a revised version of the manuscript that addresses the points raised during the review process.

We look forward to receiving your revised manuscript.

Kind regards,

Kelli K Ryckman

Academic Editor

PLOS ONE

2. Please include additional information regarding the survey or questionnaire used in the study and ensure that you have provided sufficient details that others could replicate the analyses. For instance, if you developed a questionnaire as part of this study and it is not under a copyright more restrictive than CC-BY, please include a copy, in both the original language and English, as Supporting Information. If the original language is written in non-Latin characters, for example Amharic, Chinese, or Korean, please use a file format that ensures these characters are visible.

3. Please state whether you validated the questionnaire prior to testing on study participants. Please provide details regarding the validation group within the methods section.

4. Thank you for stating the following in the Acknowledgments/Funding Section of your manuscript:

“This work was supported by the National Institutes of Health's Intramural Research Program at the *Eunice Kennedy Shriver* National Institute of Child Health and Human Development (clinical site contracts HHSN275201300013C to Northwestern University, HHSN275201300014C to the University of Alabama at Birmingham; and the Emmes Company for the Data Coordinating Center HHSN275201300026I, HHSN27500001, HHSN275000017).”

“This work was supported by the National Institutes of Health's Intramural Research Program at the Eunice Kennedy Shriver National Institute of Child Health and Human Development (clinical site contracts HHSN275201300013C to Northwestern University, HHSN275201300014C to the University of Alabama at Birmingham; and the Emmes Company for the Data Coordinating Center HHSN275201300026I, HHSN27500001, HHSN275000017).  The funders had no role in study design, data collection and analysis, decision to publish, or preparation of the manuscript.”

6. We note that you have indicated that data from this study are available upon request. PLOS only allows data to be available upon request if there are legal or ethical restrictions on sharing data publicly. For more information on unacceptable data access restrictions, please see http://journals.plos.org/plosone/s/data-availability#loc-unacceptable-data-access-restrictions.

7. PLOS requires an ORCID iD for the corresponding author in Editorial Manager on papers submitted after December 6th, 2016. Please ensure that you have an ORCID iD and that it is validated in Editorial Manager. To do this, go to ‘Update my Information’ (in the upper left-hand corner of the main menu), and click on the Fetch/Validate link next to the ORCID field. This will take you to the ORCID site and allow you to create a new iD or authenticate a pre-existing iD in Editorial Manager. Please see the following video for instructions on linking an ORCID iD to your Editorial Manager account: https://www.youtube.com/watch?v=_xcclfuvtxQ

Reviewers' comments:

Reviewer's Responses to Questions

**Comments to the Author**

1. Is the manuscript technically sound, and do the data support the conclusions?

Reviewer #1: Yes

Reviewer #2: Yes

2. Has the statistical analysis been performed appropriately and rigorously? 

Reviewer #1: Yes

Reviewer #2: Yes

3. Have the authors made all data underlying the findings in their manuscript fully available?

Reviewer #1: Yes

Reviewer #2: Yes

4. Is the manuscript presented in an intelligible fashion and written in standard English?

Reviewer #1: Yes

Reviewer #2: Yes

5. Review Comments to the Author

Reviewer #1: This is a well written paper, and novel study examining body composition and gestational weight gain and their relationship with asthma control during pregnancy.

What was the rationale for examining FEV6?

Reviewer #2: This is a prospective cohort of 299 women examining the associations between body composition and weight gain with asthma during pregnancy. The manuscript is well written and shows that increased weight and exessive weight gain in the first trimester were associated woth lung function during pregnancy.

I only have some minor comments.

Materials and Methods

In regards to the population; did you account for women twins or triplets? Were they included in the cohort?

In the last paragraph on page 7 (rows 145-148) you state that women reported asthma attacks and medical encounters from the past year; did you also have medical records to confirm this or was it just by participants recall?

Results

Do you have any explanation for the improved PEF in women with exessive weight gain? (Table S3)

Dicsussion

The FEV1/FVC seems to be unaltered during pregnancy regardless of weight gain. Still, women tend to have more asthma symptoms with increased weight. Do you have any comments on this? Is the symptomology reflecting asthma? Or something else?

The conclusion is not very clear. Now it includes parts that should be in the introduction (background) as well as parts that should be in the first paragraph of the discussion. I suggest you leave those things out and instead include something more similar to what you have in the abstract.

6. PLOS authors have the option to publish the peer review history of their article (what does this mean?). If published, this will include your full peer review and any attached files.

Reviewer #1: No

Reviewer #2: No

---

## [Author Response · Author response to Decision Letter 0]

25 Mar 2022

RESPONSE TO REVIEWERS

Reviewer #1: This is a well written paper, and novel study examining body composition and gestational weight gain and their relationship with asthma control during pregnancy.

What was the rationale for examining FEV6?

Response: FEV6 was included as part of the original study protocol based on prior studies in pregnant populations finding FEV6 to be highly correlated with FVC whilst being easier, more achievable, more reproducible, and less physically demanding1. Its inclusion in the analysis was to completely represent all measures from spirometry.

1. Zairina E, Abramson MJ, McDonald CF, et al. A prospective cohort study of pulmonary function during pregnancy in women with and without asthma. J Asthma. 2016;53(2):155-63. doi:10.3109/02770903.2015.1080268

Reviewer #2: This is a prospective cohort of 299 women examining the associations between body composition and weight gain with asthma during pregnancy. The manuscript is well written and shows that increased weight and excessive weight gain in the first trimester were associated with lung function during pregnancy.

I only have some minor comments.

Materials and Methods

In regards to the population; did you account for women twins or triplets? Were they included in the cohort?

Response: To be eligible for study participation, participants had to have a singleton pregnancy. We have added study eligibility criteria to our supplement as S1 Table and reference this table on page 6, line 108 of the revised manuscript:

“Medical record review was used to identify potentially eligible participants, who were then screened for eligibility (S1 Table) and consent”.

In the last paragraph on page 7 (rows 145-148) you state that women reported asthma attacks and medical encounters from the past year; did you also have medical records to confirm this or was it just by participants recall?

Response: A chart abstraction was completed for participants during their time in the study and included encounters within a given hospital system. Thus, chart abstractions were unfortunately not able to detail a complete history of asthma attacks and/or medical encounters.

However, you bring up a good point about the limitations of this measure. We have thus added additional language to the limitations section (page 21, lines 389-402) on self-reported measurement error. This section now reads, 

“This study was limited by a small sample size which precluded our ability to conduct subgroup analyses by asthma phenotypes. Participant self-report of certain exposure (pre-pregnancy BMI) or outcome (symptomology, exacerbations) may be subject to measurement error. Self-reported pre-pregnancy weights were highly correlated with first measured pregnancy weights (n=343, r=0.98) as well as chart-abstracted pre-pregnancy weights (n=118, r=0.94) in our cohort, and missing data imputation for pre-pregnancy BMI allowed us to produce unbiased and efficient estimates for our associations[41]. Symptomology was self-reported in daily diaries, which have been shown to provide more sensitive estimates of asthma control than retrospective questionnaires[42]. Exacerbations – especially for the first trimester, which encompassed a one-year timeframe – were subject to recall bias. However, given the severity of asthma exacerbations and disruption that they would cause, any recall bias of exacerbations is likely limited. Finally, though we account for chronic mediation use at baseline, we did not assess how changes in medication use may have impacted our associations[43].”

Results

Do you have any explanation for the improved PEF in women with excessive weight gain? (Table S3)

Response: We agree that this is a curious finding. These associations are present only for the excessive second trimester gestational weight gain; though not significant, first and third trimester excessive gestational weight gain were associated with reduced PEF. In childhood, weight gain is associated with greater lung volumes due to lung growth via IGF-1 and other factors. In adult women, this would not make too much sense. From the pregnancy literature (which is admittedly limited), there is no strong explanation. More notable, our clinical authors do not see a reason for the finding, and we are hesitant to speculate. We have added some additional language to the discussion on page 19, lines 344-346 clarifying this:

“Unexpectedly, we observed few associations between gestational weight gain and lung function other than improved percent predicted peak flow with excessive second trimester gestational weight gain. Given the lack of biologic plausibility, this finding may be due to chance.”

Discussion

The FEV1/FVC seems to be unaltered during pregnancy regardless of weight gain. Still, women tend to have more asthma symptoms with increased weight. Do you have any comments on this? Is the symptomology reflecting asthma? Or something else?

Response: Our findings are present only among participants with asthma, suggesting that the mechanisms driving this association are asthma-specific. Further, the main symptoms influenced were activity limitation, night symptoms, and rescue inhaler use; respiratory symptoms – which may be more obviously linked to lung function – were unaffected. Several potential mechanisms may explain our findings. The means by which participants regulate weight changes (e.g., exercise, diet) may differ by asthma status, and this may be the true factor influencing symptomology in participants with asthma. Asthma may be co-occurring with weight gain-related pregnancy complications (e.g., hypertensive disorders of pregnancy) to influence symptomology. Along the lines of your suggestion, symptomology may be reflecting something other than asthma (e.g., increased perception of dyspnea and fatigue associated with pregnancy-related weight gain, which may be felt more keenly among participants with asthma). 

We have discussed lifestyle factors (page 20, lines 355-359), pregnancy-related dyspnea and fatigue (page 19, lines 345-347), and now briefly reference weight gain-related pregnancy complications on page 20, lines 358-359. 

The conclusion is not very clear. Now it includes parts that should be in the introduction (background) as well as parts that should be in the first paragraph of the discussion. I suggest you leave those things out and instead include something more similar to what you have in the abstract.

Response: We have modified the Conclusions (Page 22, lines 407-422) to now read,

“Poor asthma control affects 57-80% of patients [44] and is modified for more than half of women during pregnancy[7], making identification of risk factors a priority. In this prospective pregnancy cohort of women with and without asthma, we observed asthma-specific associations between higher pre-pregnancy BMI, early pregnancy skinfolds, and excessive first trimester gestational weight gain associated with restrictive changes in lung function and poor asthma control during pregnancy. Despite experiencing decrements in lung function and asthma control, women with an obese asthma phenotype had fewer asthma-related doctor’s visits, suggesting less active management of asthma in this high-risk group. Active asthma management is key to the control of asthma and mitigation of adverse outcomes in pregnancies complicated by asthma[6]. While prenatal care visits are unrelated to asthma, they may provide an opportunity to help pregnant persons with asthma – particularly high-risk individuals with obesity or excess first trimester gestational weight gain – achieve better asthma control."

---

## [Editor Report · Decision Letter 1]

4 Apr 2022

Maternal body composition and gestational weight gain in relation to asthma control during pregnancy

PONE-D-21-21560R1

Dear Dr. Mendola,

We’re pleased to inform you that your manuscript has been judged scientifically suitable for publication and will be formally accepted for publication once it meets all outstanding technical requirements.

Kind regards,

Kelli K Ryckman

Academic Editor

PLOS ONE
---

## [Editor Report · Acceptance letter]

7 Apr 2022

PONE-D-21-21560R1 

Maternal body composition and gestational weight gain in relation to asthma control during pregnancy 

Dear Dr. Mendola:

I'm pleased to inform you that your manuscript has been deemed suitable for publication in PLOS ONE. Congratulations! Your manuscript is now with our production department. 

Kind regards, 

on behalf of

Dr. Kelli K Ryckman 

Academic Editor

PLOS ONE